# OpenReview forum: "Activation Steering via Contrastive Causal Mediation"
_ICLR.cc/2026/Conference — ICLR 2026 Conference Desk Rejected Submission_

### Official Review · Reviewer_gaHv · 2025-10-24

**Soundness:** 2
**Presentation:** 2
**Contribution:** 2
**Rating:** 4
**Confidence:** 4

**Summary:**

The paper proposes an approach to steer models when generating multiple tokens, as opposed to a single token. To do so, the authors assemble a set of contrasting prompts that are defined by the goal of the intervention. Their method, contrastive causal mediation (CCM), enables steering with fewer attention heads and lower steering factors, allowing steering across long form text generation. They also explore how their selection of attention heads approach transfers to out-of-domain datasets and show the viability of transferring the selection on different datasets.

**Strengths:**

The authors contribute a fairly useful approach to steering over multiple tokens, that of which (to my knowledge) has little prior work in. The solution is simple and can be applied to multiple datasets. In addition, the solution is a ranking technique and can also be used across multiple attention patching techniques, showcasing its applicability.

**Weaknesses:**

1. The current contribution is rather incremental because the contribution is mostly a method to create contrastive pairs for steering. One way to make the paper less incremental is to further investigate why steering certain datasets does not generalize well, because the way the datasets are investigated leaves the reader confused about why, for instance, why global steering fails on refusal and verse (for instance, what qualities of each dataset make one less amenable for steering than the other?).
2. The results on MMLU are seemingly unmotivated and out of scope. It is unclear why MMLU results are included when the dataset as a whole does not make sense as a use case for steering (why would one like to steer with attention heads selected from any of the prior three tasks for MMLU, and why should we care?). If this is better motivated, it would be very helpful.
3. The approach does not seem too generalizable since the method requires gridsearching over a number of hyperparameters, per dataset. Is there a more generalizable way to select hyperparameters?
4. In general, there are some incomplete phrases (lines 433-434) and generally typos (lines 479-480), those of which are closer to the end of the paper/conclusion section. Also, repeated description of the held-out dataset on lines 303-305 with Sec 4.3.

**Questions:**

1. Can you explain how the LLM-as-a-judge evaluation is done? It is unclear why 1-3 is considered inaccurate, and 4/5 are mapped to discrete notions of accuracy.
2. Is there a reason why human evaluation is not done with LLMs-as-a-judge to verify alignment to humans?
3. Following Q1, is there a table showing the accuracy of the LLM-as-a-judge for the queries in Table 1? Furthermore, it is a bit unclear how this is used based on the current paper description (essentially, are the queries that are given a score of 4 or 5 essentially used for the contrastive pairs?)
4. Did you try other models for the LLM-as-a-judge, since different models will rate different statements "more" or "less" contrastive?
5. How were the OOD datasets chosen?
6. Were experiments done on non-DPO instruction-tuned models (or, even base models)? Some of the earlier instruction-tuned models may not be mid-trained or post-trained with RLHF/DPO, but it would be interesting to see how the attention head selection differ between the post-training regimes.

---

> ### Author Response · Authors · 2025-11-29
>
> Thank you for your critical observations, they helped us improve the paper!
>
> **W1 Contribution is incremental** Please see the global response (and response to reviewer AQhS) for a clear description of our main contribution.
>
> **W2 MMLU results (Section 4.3)** MMLU evaluations were done to investigate whether steering on our three tasks degraded existing capabilities of the model on MMLU (a somewhat unrelated task, with precedent in earlier steering literature[1]). This is now motivated in Section 4.3 of the revised submission. Additionally, evaluations now incorporate two stronger metrics that quantify whether existing capabilities have been hampered -- relevance and fluency scores.  This enables GCM approaches to identify model sites that can steer without hampering the model's extant ability to remain on-topic and coherent in its responses.
>
> **W3 Questionable generalization since approach requires extensive grid-searching** This remains an open problem in steering, and our paper does not attempt to solve it. We hope our extensive grid search enables researchers to more quickly identify favorable fractions of attention heads and steering factors in the future! Two trends emerge from 16,200 experiments across three models, three tasks, five localization algorithms, and three steering algorithms: (1) global steering (steering factor with $(N=1)$, $(k=1)$) and localized steering ($(N>8), (0.05 \le k \le 0.08)$) both provide consistent success; and (2) identifying optimal heads to steer is highly efficient— one of the best performing GCM approaches, attribution patching (Section 2.2.1), uses gradient approximations requiring only two forward passes and one backward pass, localizing concepts in $O(1)$).
>
> **W4 typos/errors** Thank you for your care and attention to detail! These are now fixed.
>
> **Q1/Q2** - Answered in the global response and response to `kdRW`
>
> **Q3 LM Judge evals, what is it used for? (Appendix C1, Table 3)** The LM Judge is _only_ used for evaluating post-intervention steering responses. It does not play any role in localization (Also see Fig. 2). All LM judge scores which determine our steering success rate are available in Appendix C1 for all hyper-parameter settings. Additionally average scores are available in Table 3.
>
> **Q4 Other LM judges (Appendix C2, Appendix C3, Section 3.3)** While we couldn't do this during the rebuttal period, we improve the calibration of the current LM judge evaluations by incorporating additional independent evaluations for measuring relevance and fluency. We also include a human evaluation to calibrate the LM judge.
>
> **Q5 Choice of OOD Datasets** We select OOD datasets based on use in earlier research. `Alpaca` is the standard dataset for evaluating refusal [1,2]. For sycophancy, we use the `sycophancy_for_NLP` dataset, which induces long form responses and has been used in other steering research that studied sycophancy [3]. For verse style transfer, we use the reddit writing prompts dataset [4] since the questions are open ended and not factual.
>
> **Q6 Non DPO models** Our primary criteria during model selection was strong baseline performance, which all three models achieved while being from diverse families and small or moderate in size. This is a great idea for future work. Base models may more clearly show the influence of attention heads on long-form generation given their pure auto-regressive training!
>
>
> [1] Arditi, Andy, et al. "Refusal in language models is mediated by a single direction." Advances in Neural Information Processing Systems 37 (2024): 136037-136083.
>
> [2] Zhao, Jiachen, et al. "Llms encode harmfulness and refusal separately." arXiv preprint arXiv:2507.11878 (2025).
>
> [3] Rimsky, Nina, et al. "Steering llama 2 via contrastive activation addition." Proceedings of the 62nd Annual Meeting of the Association for Computational Linguistics (Volume 1: Long Papers). 2024.
>
> [4] Fan, Angela, Mike Lewis, and Yann Dauphin. "Hierarchical neural story generation." arXiv preprint arXiv:1805.04833 (2018).

---

### Official Review · Reviewer_FEVr · 2025-10-31

**Soundness:** 4
**Presentation:** 4
**Contribution:** 3
**Rating:** 8
**Confidence:** 4

**Summary:**

This paper explores how to extract localized model components for steering models around concepts that are diffused over multiple tokens. They design a method, Contrastive Causal Mediation, that identifies subsets of attention heads that most impact model behavior on a dataset of contrastive pairs illustrating the concept to steer. The paper shows that the CCM selection methodology in numerous variants applies to multiple different state-of-the-art steering techniques, often requiring to patch only a small fraction of attention heads. They also show that patching attention heads can generalize to held-out, in-domain datasets, and showed an experiment that showed how LLMs can be steered even when performing QA on MMLU.

**Strengths:**

W1. The idea of deriving steering attention heads from causal mediation on contrastive sets is novel (to my knowledge), clever, and effective.

W2. The paper's results are substantial, and it considers numerous state-of-the-art baselines and ablations to the to selection method (CCM) and steering methods.

W3. The paper is very clear, well written, and a pleasure to read.

W4. The selection approach, which is based on relative log probs, is modular and agnostic to steering algorithm; this means it can be generally applied to future, improved methods that may develop, making the contribution likely to last longer than other methods.

**Weaknesses:**

W1. The CCM approach is illustrated on relatively short prompts; it would be beneficial for the paper to show results on tasks with prompts that are nontrivially long, matching many realistic task settings.

W2. (minor) The results for OlMo and SOLAR are slightly worse than for Qwen, but the grid search still generally achieves adequate steerability rates

**Questions:**

1. In Figure 4, what explains the high varaibility in OLMo performance with local steering on the refusal task?

2. a couple of \cite{} should be \citep{},e.g. L.360 (REFT Wu et al), 427 Park et al

3. Sentence ends prematurely on l479 - "and single-token responses, and how do they Using"

---

> ### Author Response · Authors · 2025-11-29
>
> Thanks a lot for your kind encouragement!
>
> **W1. Results are illustrated on short prompts** - This is a great idea for future work. We work with responses that are \~128 tokens long and will explore whether localization signals become weaker/stronger as response length increases/decreases.
>
> **W2. The results for OlMo and SOLAR are slightly worse (Appendix C1)** - This is a good observation. Model performance varies across tasks: Qwen and OLMo perform better than SOLAR on refusal induction, whereas SOLAR excels on verse style transfer. Overall, our grid search indicates that unsupervised steering works best with large steering factors and small top-$k$ values for localized steering, and with small steering factors and large top-$k$ values for effective global steering. For ReFT, a normalized, unscaled steering vector applied to a small fraction of attention heads yields strong results.
>
> **Q1 High variability in OLMo performance with local steering... (Figure 3)**  As noted in the global response, this was because of a bug in an upstream package. The fixed results do not show high variance.
>
> **Q(all) nits** Thank you for your care and attention to detail. These are now fixed!

---

### Official Review · Reviewer_AQhS · 2025-10-31

**Soundness:** 2
**Presentation:** 2
**Contribution:** 2
**Rating:** 4
**Confidence:** 3

**Summary:**

This paper proposes a method Contrastive Causal Mediation for selecting which attention heads to intervene on when attempting to steer for a particular concept or behavior. The method is agnostic as to the actual form of intervention, it just tells you which attention heads are the most important. Specifically the method ranks attention head importance by measuring the causal effect of each head on inducing the behavior when intervened upon. The paper tests the efficacy of the method across three tasks: refusal induction, sycophancy, and a "talk in verse" style transfer task. They compare against random baselines and other techniques such as probing for the concept using head activations.

**Strengths:**

The paper tests a range of hyperparameters for steering such as steering magnitude and fraction of attention heads for intervention. The paper evaluates their method across three distinct settings: style transfer, refusal, sycophancy. They also test for capabilities preservation by evaluating against MMLU.

**Weaknesses:**

I feel the main weakness of this paper is that its core finding - that testing the causal effect of intervening on attention heads is an effective method for selecting intervention sites - is not novel: Function Vectors in Large Language Models (https://arxiv.org/pdf/2310.15213) (Todd 2024) introduces a "causal indirect effect" score for ranking attention head importance as intervention sites (equation 3). Their score and the "indirect effect" score in this paper (line 165) are largely the same.

**Questions:**

Nit: for figure 2 I guess the final columns of each graph per row should be identical? There are small differences for sycophancy - maybe would make sense to separate out the final column visually.

A question about interpreting the efficacy of steering for the verse task - would it be that the capability is localized or distributed? It seems only by intervening on all heads does it work to induce the model to speak in verse.

---

> ### Author Response · Authors · 2025-11-29
>
> Thank you for taking the time to review our paper!
>
> **W1. Novelty/Main contribution** - We clarify our main contributions in the global response. Our primary contribution is showing that realistic, long-form generations contain rich concept signals that support both localization and steering—addressing evaluation desiderata [1] unmet by prior steering work. A second contribution is our comprehensive, systematic study of mediation-guided steering location selection. Although attention-head analyses have precedent predating Todd et al. [2–4], our work extends these analyses to localization using signals from long-form responses and evaluates them comprehensively and at scale.
>
> **Q1. Nit (Fig. 2)** -- thanks for catching our bug. This is now fixed in the revised submission.
>
> **Q2. Efficacy of steering in the verse style transfer task (Figures 6-9)** - Great question! I have three observations here: (1) For Qwen and SOLAR, verse style transfer localizes well: steering <5% of attention heads yields >80% success, and OLMo reaches similar success by steering ~8% of heads. (2) Verse style transfer is our most challenging task. Unlike refusal induction -- with clear lexical markers such as “I’m sorry,” “As an AI,” and “I cannot” [5,6] -- or sycophancy, which often has distinctive openings, verse has no short token span that reliably characterizes the target style. Models produce diverse poetic forms (haiku, iambic pentameter, etc.), making the signal more diffuse. (3) Our localization uses univariate causal models (Appendix B1), but verse-transfer likely involves multiple mechanisms. Localization may isolate only one, while global steering may leverage several. As supporting evidence, certain random-head configurations (e.g., steering 6% of heads with factor 10) achieve up to 98% success (Fig. 6).
>
> ### References
> [1] Pres, Itamar, et al. "Towards reliable evaluation of behavior steering interventions in llms." arXiv preprint arXiv:2410.17245 (2024).
>
> [2] Li, Kenneth, et al. "Inference-time intervention: Eliciting truthful answers from a language model." Advances in Neural Information Processing Systems 36 (2023): 41451-41530.
>
> [3] Wang, Kevin, et al. "Interpretability in the wild: a circuit for indirect object identification in gpt-2 small." arXiv preprint arXiv:2211.00593 (2022).
>
> [4] Todd, Eric, et al. "Function vectors in large language models." arXiv preprint arXiv:2310.15213 (2023).
>
> [5] Mazeika, Mantas, et al. "Harmbench: A standardized evaluation framework for automated red teaming and robust refusal." arXiv preprint arXiv:2402.04249 (2024).
>
> [6] Arditi, Andy, et al. "Refusal in language models is mediated by a single direction." Advances in Neural Information Processing Systems 37 (2024): 136037-136083.

---

### Official Review · Reviewer_kdRW · 2025-10-31

**Soundness:** 2
**Presentation:** 2
**Contribution:** 3
**Rating:** 4
**Confidence:** 3

**Summary:**

The paper introduces a new method to select attention heads for steering, particularly useful long-form generation. The method ranks the attention heads by identifying which ones are associated with a higher probability of a contrastive response w.r.t. the original response. The authors claim the method is more efficient than a random baseline and standard probing.

**Strengths:**

- Novel method for steering in long-form generation by identifying which attention heads increase the probability of contrastive pairs.
- Evaluation was conducted on held-in and held-out datasets.

**Weaknesses:**

- LLM as a judge set up:
    - Unclear scale: First, the model is asked to come up with a Likert scale 1-5, then 1-3 is mapped to 0, 4 to 80 and 5 to 100 to get a score. Any reason why not just ask LLM to directly map to the 3 categories?
    - Lack of validation: No validation of LLM as a judge is mentioned (particularly it is not clear how random or purposeful are 80 and 100 distinctions or 4/5 likert scale distinctions).

- Experimental set up: the experimental set up does not seem to align with the central claim of the paper. The central claim of the paper is that "CCM is consistently better than correlational baselines that use probes to select attention heads for steering." To show this, the authors need to show that across models, steering methods and for best hyperparameters CCM outperforms Inference-time-interventions and Random selections (at least -- these are the baselines identified by the authors). However, in their actual set up only includes difference-in-means steering to compare the effectiveness of CCM and ITI. It seems 4.2 only compares the steering methods themselves, and it is unclear whether CCM was best for each of these methods.

- Analysis and presentation of results: the main results of the paper are supported by Figure 2, which presents a multitude of heatmaps that are difficult to parse. For example, to support their claim that CCM variants are more efficient than probing, the authors vaguely refer to "upper left corner" of CCM heatmaps. However, depending on how this corner is defined, it appears only the sycophancy task has a somewhat noticeable difference, while others are pretty much comparable. Moreover, there is simply no notion of statistical significance presented in the results. There were no tests or p values, no confidence intervals, and hence all of these trivial differences may have just occurred due to noise. Visually it does seem that sometimes the CCM selection is more efficient, but the authors need to devise a more rigorous procedure to test that, as currently they rely mostly on exploratory analysis.

**Questions:**

nitpicks:
- 2.2 and 2.3 should be joined in one section
- Table 1 refers to ratings as Likert scales but they seem more like preference or concept detection prompts. Later it is described that they request LLMs to output on a scale but it was confusing initially.
- The formula on line 256 is a bit out of context and not explained, could probably be removed
- line 297 space before comma
- section 2.1 and Tasks in Section 3 seem a bit repetitive. It is also confusing what overall was the pipeline for data generation. These sections need significant revisions for clarity. For example, distinction between base and source queries is not introduced and hence very confusing.
- not clear what "strict" means in line 322. Also this is redundant with lines 306-310 just above.
- figrue 2 appears before it is referred to

---

> ### Author Response · Authors · 2025-11-29
>
> We thank the reviewer for their sharp observations. Your review helped us improve our paper!
>
> We restate a few points from the global response here for convenience. The revised submission also includes expanded tables and additional appendices.
>
> **W1. LM as a judge (Appendix C2, Section 3.3)** - The original submission rescaled Likert ratings to distinguish 4- and 5-point scales, ensuring conservative estimates. LM-judge scores for the three concepts are now binarized: only responses with maximum scores are accepted. Additional relevance and fluency judges are included, and intervened responses must achieve top scores on all three (concept, relevance, fluency) axes to be accepted.
>
> **W2/W3. Experimental set-up/presentation of results (Appendix C1, Table 3)** - The original submission evaluated GCM primarily with difference-in-means steering. We now provide a full hyperparameter search for all steering strategies—16,200 experiments with 810,000 repeated measurements—giving a complete picture of GCM performance. Heatmaps in Fig. 2 and Figs. 6–13 are summarized in Table 3 for easier interpretation.
>
> **W3. Statistical tests (Appendix C1.5)** - All results are now supported by statistical tests. GCM variants, Activation and attribution patching significantly outperform ITI and random baselines (p < 0.001), while attention-head knockouts exceed random selection but not ITI. Across tasks and models, GCM variants outperform baselines in 78–95% of settings under unsupervised steering (p < 0.05), and match baseline performance under supervised steering (p > 0.05).
>
> **Q (all)  Nits** Thank you for your care and attention to detail. We have updated the revised manuscript with all suggested changes.

---

### Author Response · Authors · 2025-11-29
**Global rebuttal**

We thank all reviewers for their thoughtful feedback and kind attention to our work.  We have substantially revised the paper with consideration to reviewer suggestion.

We would like to clarify our main contributions (Reviewers `gaHv`, `AQHs`).

## Contributions

**Localization in a generative-response setting.**   The paper tackles the underexplored problem of localizing concepts in LLMs using signals from long-form generative outputs (~128 tokens). Prior interpretability and causal mediation studies [1–9] focus largely on single-token or synthetic tasks, overlooking concepts expressed over many tokens (e.g., verse vs. prose). We demonstrate that generative responses offer rich, realistic signals for concept localization. To emphasize this, we rename our method **Generative Causal Mediation (GCM)** and update the paper accordingly.

**Comprehensive evaluation.**  The study spans 5,400 hyperparameter settings and 270,000 repeated measurements (expanded to 16,200 and 810,000 during rebuttal) across 3 models, 3 tasks, 5 localization methods, and 3 steering techniques. GCM outperforms linear probes and random baselines under unsupervised steering (difference-in-means and means), and performs comparably to baselines under supervised steering (ReFT).

**Localization is helpful but not required for model control.**  After correcting a bug from the original submission, we find that both localized and global steering (N=1, top-k=1) transfer well to held-out datasets. Localization remains valuable for mechanistic insight but is not essential for effective model control.

## Rebuttal

- **Expanded evaluations (Appendix C1, Table 3)** (Reviewer `kdRW`).  The original submission evaluated GCM primarily with difference-in-means steering. We now provide a full hyperparameter search for all steering strategies—16,200 experiments with 810,000 repeated measurements—giving a complete picture of GCM performance. Heatmaps in Fig. 2 and Figs. 6–13 are summarized in Table 3 for easier interpretation.
- **Statistical analysis (Appendix C1.5)** (Reviewer `kdRW`).  Activation and attribution patching significantly outperform ITI and random baselines (p < 0.001), while attention-head knockouts exceed random selection but not ITI. Across tasks and models, GCM variants outperform baselines in 78–95% of settings under unsupervised steering (p < 0.05), and match baseline performance under supervised steering (p > 0.05).

- **Improved and calibrated LM judge (Appendix C2, Section 3.3)** (Reviewers `kdRW`, `gaHv`).  LM-judge scores for the three concepts are now binarized: only responses with maximum scores are accepted. Additional relevance and fluency judges are included, and intervened responses must achieve top scores on all three (concept, relevance, fluency) axes to be accepted.
- **Human calibration (Appendix C3)** (Reviewers `kdRW`, `gaHv`).  We calibrate the LM judge using a uniformly stratified sample of 1,000 prompts and model responses across all five (3 concepts, relevance, fluency) axes. Model–human agreement ranges from 0.82 to 0.95 (macro-average 0.87), with Cohen’s $\kappa$ indicating substantial agreement.

---

> ### Author Response · Authors · 2025-11-29
>
> [1] Wang, Kevin, et al. "Interpretability in the wild: a circuit for indirect object identification in gpt-2 small." arXiv preprint arXiv:2211.00593 (2022).
>
> [2] Todd, Eric, et al. "Function vectors in large language models." arXiv preprint arXiv:2310.15213 (2023).
>
> [3] Vig, Jesse, et al. "Investigating gender bias in language models using causal mediation analysis." Advances in neural information processing systems 33 (2020): 12388-12401.
>
> [4] Finlayson, Matthew, et al. "Causal analysis of syntactic agreement mechanisms in neural language models." Proceedings of the 59th Annual Meeting of the Association for Computational Linguistics and the 11th International Joint Conference on Natural Language Processing (Volume 1: Long Papers). 2021.
>
> [5] Sankaranarayanan, Aruna, Dylan Hadfield-Menell, and Aaron Mueller. "Disjoint processing mechanisms of hierarchical and linear grammars in large language models." arXiv preprint arXiv:2501.08618 (2025).
>
> [6] Prakash, Nikhil, et al. "Language models use lookbacks to track beliefs." arXiv preprint arXiv:2505.14685 (2025).
>
> [7] Sharma, Arnab Sen, et al. "LLMs Process Lists With General Filter Heads." arXiv preprint arXiv:2510.26784 (2025).
>
> [8] Gur-Arieh, Yoav, Mor Geva, and Atticus Geiger. "Mixing Mechanisms: How Language Models Retrieve Bound Entities In-Context." arXiv preprint arXiv:2510.06182 (2025).
>
> [9] Rimsky, Nina, et al. "Steering llama 2 via contrastive activation addition." Proceedings of the 62nd Annual Meeting of the Association for Computational Linguistics (Volume 1: Long Papers). 2024.

---

### Author Response · Authors · 2025-12-02

Dear AC,

We wanted to bring to your attention that the ICLR review period was frozen before our reviewers had the chance to respond to our revisions. We're grateful to all the reviewers for their feedback, it hugely improved our original submission.

Given this new situation, we would be grateful if you could take a moment to carefully review our updated submission and rebuttal. In our revision, we addressed all points raised by all reviewers. We also substantially expanded our study. Our evaluations are now more extensive, and include a human calibration of our LLM judge model and an exhaustive set of hyper-parameter explorations. The results support our original claims.

We have uploaded a revised camera-ready version that reflects all of these improvements.

Thank you very much for your time and consideration!

---

### Note · Program_Chairs · 2026-01-17
**Submission Desk Rejected by Program Chairs**

The following references in this submission do not refer to real documents and/or have major errors in bibliographic information:

 - Piotr Stanczak and Yonatan Belinkov. A causal framework for discovering and removing gender bias in language representations. arXiv preprint arXiv:2210.06817, 2022.
- Prakhar Dathathri, Andrea Madotto, Zhaojiang Lan, Jamin Hung, Ehsan Frank, Jason Liu, and Pascale Fung. Plug and play language models: A simple approach to controlled text generation. In International Conference on Learning Representations, 2020.
- X. Li, Y. Zhang, and P. Wang. Activation editing for steering language models. arXiv preprint arXiv:2308.10248, 2023b.
- Julian Michael, Alex Warstadt, and Ellie Pavlick. Causal mediation analysis of syntactic agreement in large language models. arXiv preprint arXiv:2311.09898, 2023.
- Yukai Zhou, Zhijie Huang, Feiyang Lu, Zhan Qin, and Wenjie Wang. Don’t say no: Jailbreaking llm by suppressing refusal. arXiv preprint arXiv:2405.00049, 2024.